# CXCL14 as a Key Regulator of Neuronal Development: Insights from Its Receptor and Multi-Omics Analysis

**DOI:** 10.3390/ijms25031651

**Published:** 2024-01-29

**Authors:** Yinjie Zhang, Yue Jin, Jingjing Li, Yan Yan, Ting Wang, Xuanlin Wang, Zhenyu Li, Xuemei Qin

**Affiliations:** 1Modern Research Center for Traditional Chinese Medicine, The Key Laboratory of Chemical Biology and Molecular Engineering of Ministry of Education, Shanxi University, No. 92, Wucheng Road, Taiyuan 030006, China18235431334@163.com (T.W.);; 2Engineering Research Center of Cell & Therapeutic Antibody, School of Pharmacy, Shanghai Jiao Tong University, Shanghai 200240, China

**Keywords:** *CXCL14*, metabolomics, neuron development, MALAR-Y2H

## Abstract

*CXCL14* is not only involved in the immune process but is also closely related to neurodevelopment according to its molecular evolution. However, what role it plays in neurodevelopment remains unclear. In the present research, we found that, by crossbreeding CXCL14^+/−^ and CXCL14^−/−^ mice, the number of CXCL14^−/−^ mice in their offspring was lower than the Mendelian frequency; CXCL14^−/−^ mice had significantly fewer neurons in the external pyramidal layer of cortex than CXCL14^+/−^ mice; and *CXCL14* may be involved in synaptic plasticity, neuron projection, and chemical synaptic transmission based on analysis of *human* clinical transcriptome data. The expression of *CXCL14* was highest at day 14.5 in the embryonic phase and after birth in the mRNA and protein levels. Therefore, we hypothesized that *CXCL14* promotes the development of neurons in the somatic layer of the pyramidal cells of mice cortex on embryonic day 14.5. In order to further explore its mechanism, CXCR4 and CXCR7 were suggested as receptors by Membrane-Anchored Ligand and Receptor Yeast Two-Hybrid technology. Through metabolomic techniques, we inferred that *CXCL14* promotes the development of neurons by regulating fatty acid anabolism and glycerophospholipid anabolism.

## 1. Introduction

C-X-C motif chemokine ligand 14 (*CXCL14*), also known as BRAK, is a non-ELR (glutamic acid-leucine-arginine) chemokine with a broad spectrum of biological activities [1]. The chemokine (C-X-C Motif) ligand 14 gene (also known as *BRAK*, *BMAC*, or *Mip-2γ*), located on human chromosome 5q31, is expressed as a 99 amino acid residue precursor protein, which is processed to a 77 amino acid mature protein with a molecular weight of 9.4 kDa, having a highly basic isoelectric point of 9.9 [2]. It is the evolutionary ancient chemokine for a highly conserved sequence and its homeostatic roles, which suggests that *CXCL14* has important functions in growth and development [3]. *CXCL14* plays important roles in immune surveillance, inflammation and tumor development by regulating cell migration [4].

Previous studies showed that *CXCL14* is expressed at high levels in fetal, adult and adult mouse brain tissues, including the cortex, hippocampus and cerebellum. A recent study indicated that *CXCL14* regulates the development of the microglia in the early cortex, and it is important for cell localization in early cortical stages [5]. *CXCL14* is expressed in GABAergic interneurons of the adult mouse dentate gyrus and inhibits GABAergic transmission to neural stem cells [6,7]. In stark contrast, *CXCL12* was shown to enhance the activity of gamma-aminobutyric acid (GABA) released by synapses [6]. Moreover, recent evidence revealed that *CXCL14* participates in glucose metabolism, feeding behaviour-associated neuronal circuits, and anti-microbial defense [4,8].

The activation or blocking of receptors is the main way for chemokines to exert their biological functions. Although scattered reports argued that the receptor for *CXCL14* may be CXCR4 or CXCR7, it is still not clear. Some researchers have argued that *CXCL14* could influence CXCL12–CXCR4-mediated signals [9,10]. Other studies have concluded that *CXCL14* is not a direct modulator of CXCR4 [11]. Therefore, exploring the receptor of *CXCL14* is necessary to reveal its function [10]. Membrane-Anchored Ligand and Receptor Yeast Two-Hybrid (MALAR-Y2H) is a self-developed technique for detecting interactions of ligands and membrane receptors [12]. This technique facilitates receptor identification [13].

Metabolomics systematically and comprehensively analyzes metabolites in living organisms and compare changes in metabolic networks under different physiological states and finally identifies characteristic metabolic effects associated with phenotypes. Metabolomics helps to elucidate the effects of *CXCL14* on neurodevelopment in mice by comprehensively analyzing changes in metabolites.

This study comprehensively reveals the influence of *CXCL14* on neural development from macro to micro. Firstly, the difference in birth rate, death rate and neuron density of CXCL14^−/−^ and CXCL14^+/−^ mice were studied. Secondly, the expression difference of *CXCL14* in major organs of mice and different embryonic development times was determined by quantitative PCR (Q-PCR). The interaction between *CXCL14* and its receptor was then determined based on the MALAR-Y2H system. Finally, the metabolomics of CXCL14^−/−^ and CXCL14^+/−^ were studied. By comprehensively analyzing the changes of metabolites in vivo, it is helpful to elucidate the effect of *CXCL14* on neurodevelopment in mice. In summary, we investigated the role of *CXCL14* in development by crossbreeding CXCL14^+/−^ and CXCL14^−/−^ mice. We confirmed its expression timing and location using real-time quantitative PCR (Q-PCR) and Western blot, detected its receptors through MALAR-Y2H, and speculated on its mechanism using metabolomic techniques.

## 2. Results

### 2.1. CXCL14 Knockout Caused the Reduction of Pyramidal Cells in the External Pyramidal Layer and Led to Death in Embryonic Mice

To investigate the effect on the birth rate of *CXCL14* knockout, we studied the genotype ratio of the offspring of crossbred 8-week-old male CXCL14^−/−^ and female CXCL14^+/−^ mice, which produced a total of 59 mice: 42 mice of CXCL14^+/−^, and 17 mice of CXCL14^−/−^. The results, as shown in Figure 1A, show that intercrossing yielded a ratio of 28.8% CXCL14^−/−^ to 71.2% CXCL14^+/−^ mice rather than the predicted Mendelian frequency of 50:50.

In order to explore the effect of *CXCL14* knockout on mice brains, slices of the parietal cortex of the brain of *CXCL*14^−/−^ and CXCL14^+/−^ mice were observed by Nissl staining. The results showed that the number of pyramidal cells in CXCL14^−/−^ mice was significantly lower than that in CXCL14^+/−^ mice of pyramidal cells in the external pyramidal layer (Figure 1B,C). These results suggest that *CXCL14* has an effect on the cerebral cortex development of mice.

### 2.2. The Expression of CXCL14 in Spatial and Temporal

The expression of *CXCL14* in the main organs (small intestine, brain, heart, kidney, spleen, stomach, liver, skin, lung, testis and ovary) from 8-week-old mice (*n* = 3) was detected by real-time quantitative PCR. The results showed that *CXCL14* was highly expressed not only in the skin, but also in the brain and lung (Figure 2A). 

The neuron density in each layer of the cerebral cortex was analyzed in 8-week-old CXCL14^−/−^ and CXCL14^+/−^ mice (*n* = 3). Cerebral cortex parietal lobe tissues were collected at various embryonic stages (E10.5, E12.5, E14.5, E16.5, E18.5, E19.5), as well as at postnatal days 1, 14, and 56 (P1, P14, P56). The expression level of *CXCL14* in the cerebral cortex was determined using quantitative PCR and Western blot analysis (*n* = 3).The results of both tests showed that the expression of *CXCL14* began at embryonic day 10.5, reached a peak at embryonic day 14.5, and then gradually decreased and maintained a certain expression level (Figure 2B,C). The abnormal death time of *CXCL14* knockout mice in the embryonic period was concentrated in the middle embryonic period (E10.5~E14.5), which was the most active period of cerebral cortex and central nervous system development. During this period, *CXCL14* showed a high level of expression in the cerebral cortex, suggesting that *CXCL14* was related to cortical neural development in embryonic mice.

### 2.3. CXCL14 Play Important Functions in the Cerebral Cortex

The top 1000 genes that have the most similar expression pattern to *CXCL14* in cerebral cortex were mined by the “Similar Genes Detection” function in the GEPIA website from cerebral cortex dataset of GTEx database. The genes were ranked by Pearson’s correlation coefficient (PCC). These genes were enriched in the anchoring junction, synapse, neuron projection, chemical synaptic transmission, and dendrite-related pathways (Figure 3A). The key genes related to neuron projection and dendrite, SLC6A1, GPM6A, RELN, CALB showed strong correlations with *CXCL14* (Figure 3B, *p* < 0.001).

### 2.4. There Is a Clear Interaction between CXCL14 and CXCR4 and CXCR7

Membrane-Anchored Ligand and Receptor Yeast Two-Hybrid (MALAR-Y2H) is a novel method for detecting the interaction between ligands and receptors. This method uses transmembrane peptide to anchor the ligand and the C-terminus of ubiquitin (*Ubi4*) residue *C_ub_* to the yeast cell membrane, while the other half of the ubiquitin residue *N_ub_ G* is fused with the ligand and expressed on the cell membrane. When the ligand and receptor interact, the intracellular split *C_ub_* and *N_ub_G* reconstitute as a whole and activate ubiquitin that is recognized by ubiquitin-specific proteases (*UBPs*) and released GAL4 from the C-terminal of *C_ub_*, which triggers transcription of reporter genes (Figure 4A).

The interaction between CXCL14 and CXCRn showed that CXCL14 interacted with CXCR4 and CXCR7 (Figure 4C), suggesting that the receptor of CXCL14 may be CXCR4 and CXCR7.

### 2.5. Multivariate Data Analysis from UPLC-MS

Ten serum samples were collected from both genotypic mice of CXCL14^+/−^ (W) and CXCL14^−/−^ (C). To enhance the analysis of ultra-performance liquid chromatography/tandem mass spectrometry (UPLC-MS, LC-MS) data, the raw UPLC-HRMS data were imported into LC/MS and GC/MS Data Analysis (XCMS 3.4) software for peak data matching and alignment. Subsequently, 7252 peaks were retained (Appendix A). A total of 7252 peaks were imported into SIMCA-P 14.1 (Umetrics, Umea, Sweden) for multivariate statistical analysis, including principal components analysis (PCA), and orthogonal partial least-squares discriminant analysis (OPLS-DA). In Figure 5A, principal component analysis (PCA) showed that there was a tendency for intergroup separation.

The Orthogonal Partial Least Squares-Discriminant Analysis (OPLS-DA) model established by positive ion mode data, R2, Q2 ≥ 0.5, is stable and reliable in Figure 5B,C (R2: represents the model interpretation rate; Q2: represents the predictive ability of the model; The closer R2 and Q2 are to 1, the more stable the model is Reliable.). With the OPLS-DA model VIP > 1 and *p* value < 0.05 as screening criteria, significant differential metabolites were screened. The variable importance in the projection (VIP) statistic of the first principal component of the OPLS-DA model (threshold > 1), coupled with the *p*-value of Student’s *t*-test (threshold < 0.05), was used for selecting significant variables responsible for group separation. The results showed there were a total of 50 differential metabolites between W and C groups (Figure 5D).

### 2.6. Analysis of Endogenous Metabolites and Pathway

The differences in peak areas of several metabolites were associated with CXCL14 knockout. The content of palmitic acid (PA), alpha-linolenic acid (ALA) and linoleic acid in the serum of CXCL14^−/−^ mice is lower than that in CXCL14^+/−^ mice. The pathway analysis of differential metabolites was conducted using MetaboAnalyst 4.0 software. The color and size of circles depended on the *p* value and pathway impact value analyzed by MetaboAnalyst 4.0, respectively. The bigger and the closer to the top right corner of the plot represented the more important metabolites. Therefore, we have obtained the following important metabolic pathways of linoleic acid metabolism, taurine and hypotaurine metabolism, alpha-linolenic acid metabolism, glycerophospholipid metabolism, nicotinate and nicotinamide metabolism.

## 3. Discussion

*CXCL14* is a chemotactic cytokine belonging to a small molecule-secreted protein superfamily. Studies showed that it mainly regulates cell migration, which plays an important role in immune surveillance, inflammation and cancer [14,15].

However, in research on the molecular evolution of the CXC chemokine family, the CXC chemokine family was formed before the emergence of vertebrates [16]. Four branches were generated through two genome duplications and developed into genes common to vertebrates-*CXCL14* and *CXCL12* [17]. Based on the timing of the two genomic repetition events, chemokines appear during the formation of the higher nervous system before the formation of the higher immune system. That suggests that ancestral genes in this family were initially involved in the genesis and development of the nervous system. It was previously thought that the chemokine family mainly plays the role of chemotaxis of immune cells, but *CXCL14* seems to be an exception, which does not participate in the regulation of the innate immune response of immune cells [3]. So it is speculated that *CXCL14* has always maintained the neurodevelopmental functions of its ancestral genes, but has not further acquired other functions.

The research showed that the birth rate of CXCL14^−/−^ mice was about half the expected Mendelian ratio (Figure 1A) [18]. As CXCL14^−/−^ female mice mated with CXCL14^−/−^ male mice, pregnancy characteristics and abdominal bulge can be observed on days 10–15 after mating. Seven of 10 female mice stopped gestation after 10–15 days of mating. One of the mice was unable to conceive (Appendix A). Therefore, *CXCL14* gene knockout is the main cause of mouse embryonic death, which is presumed to be related to development. In view of its evolutionary characteristics in neurodevelopment, on the one hand, CXCL14^−/−^ mice had significantly fewer neurons in layer III of the cerebral cortex (external pyramidal layer) than CXCL14^−/+^ mice by Nissle staining (Figure 1B,C). On the other hand, the analysis of *CXCL14* expression in various organs of mice also indicated that *CXCL14* expression was higher in the brain (Figure 2A). These two pieces of evidence confirmed that *CXCL14* was involved in the development of neurons.

In order to explore the influence of *CXCL14* on the brain development phase, we focused on *CXCL14* expression on the cerebral cortex parietal lobe tissue of mice at different embryonic development times by Q-PCR and Western blot. The result showed that the expression of *CXCL14* increased significantly at E14.5 during embryonic development. This inferred that *CXCL14* plays an important role in brain development around E14.5 period, which meant the death of *CXCL14* knockout mice is likely to be concentrated in this period (Figure 2B,C and Appendix A).

The neocortex develops from L4 to L3 in an inside-first, outside-last principle concentrated on E14-E15 [19]. In early development before neurogenesis, progenitor cells are found in the ventricular zone of the neural tube. The subventricular zone continues to divide and multiply. Then, after the completion of division, these immature neurons migrate radially to the appropriate parts of the cortex. Starting from E10.5 in fetal mice, the generated cortical neurons first form the cortical layer and subsequently migrate over. New neurons divide the preplate layer into the superficial marginal zone and the deeper subplate [20]. With the continuous development of neurogenesis, neuronal precursor cells undergo asymmetric division and differentiation to produce various types of projection neurons on E11.5 [21]. The neurons in the deeper subplate region formed first, followed by layer (L6, L5) neurons on E12. Eventually, upper layer (L4, L3, L2) neurons are formed during E13–E16 [22]. Therefore, according to our results, *CXCL14* is involved in the development of the cerebral cortex of E14–E15 from L4 to L3, which results in the development of neurons in the somatic layer of the pyramidal cells. When *CXCL14* of mice was knocked out, it caused abnormal development of neurons in the external pyramidal layer of the pyramidal cells.

The biological function of similar expression pattern genes with *CXCL14* in the cerebral cortex was analyzed by GO analysis. The correlated pathways, such as anchoring junction, synapse formation, neuron projection, chemical synaptic transmission, and dendrite formation, were necessary stages in the neuron development process (Figure 3A,B). This suggests that the function of *CXCL14* may be involved in one or more of these processes. Of particular note is the calcium-binding protein Calbindin 2(Calb2), an early developmental marker of midbrain dopamine cells. The same as *CXCL14*, it was the highest expression on embryonic day 14.5 (E14.5).

The absence of *CXCL14* caused the reduction of neurons in the somatic external pyramidal layer, which was probably related to the migration and differentiation function of neurons. In the nervous system, chemokines and their receptors are not only inductively expressed in inflammation, but also constitutively expressed in physiological situations. It plays an important regulatory role in the differentiation, migration, apoptosis, proliferation and activation of neurons.

CXCL14 interacted with CXCR4 and CXCR7 in experiments of MALAR-Y2H (Figure 4B,C). Coincidentally, CXCR4 and CXCR7 also are receptors of CXCL12, whose function has been confirmed in the neuro-chemotactic field [23]. The mainstream view considers that CXCR7 can regulate the function of CXCR4 by forming heteropolymers with CXCR4 [24]. It has been suggested that CXCL14 was a natural inhibitor of the CXCL12-CXCR4 signal axis by interacting with CXCR4 [25]. CXCR4 was also highly expressed during embryonic brain development. It could trigger the PI3K/AKT signaling pathway, which regulates cell proliferation and differentiation by activating D-cyclins [26,27]. Other evidence suggests that CXCR4 activation caused the MAPK signaling pathway to activate the ERK1/2 and JAK/STAT, promoting the growth and differentiation of neurons [28,29,30,31]. In this experiment, the number of cortical neurons decreased at E13.5, while *CXCL12* knockout mice showed abnormal aggregation of neurons from E13-E14 in the embryonic stage.

Metabolomics was used to explore the mechanism of *CXCL14* on neuronal growth. Most of the differential metabolites pointed to the pathway of glycerophospholipid anabolism and fatty acid anabolism, in which *CXCL14* played a particularly significant role in regulating alpha-linolenic acid and linoleic acid (Figure 5A–C).

Polyunsaturated fatty acids (PUFAs) are divided into two types of unsaturated fatty acids, one is the Ω-3 (Omega-3) series of polyunsaturated fatty acids based on alpha-linolenic Acid (ALA), which can be metabolized into Docosahexaenoic Acid (DHA); The other is the Ω-6 (Omega-6) series of polyunsaturated fatty acids based on linoleic acid, which can synthesize gamma-linolenic acid and arachidonic acid (Figure 6E) [32,33]. During embryonic development, non-esterified polyunsaturated fatty acids can bind to the ligand domain of nuclear receptors and promote neurogenesis, neuronal differentiation and synaptic plasticity [34,35]. In addition, arachidonic acid and docosahexaenoic acid are the main long-chain polyunsaturated fatty acids (LCPUFA) in the membranes of brain cells, which can increase the length and number of branches of synapses [36,37,38]. The levels of Palmitic acid, linoleic acid and alpha-linolenic acid in the serum of CXCL14^−/−^ mice were significantly lower than those in CXCL14^+/−^ mice (Figure 6A–C). The results suggested that *CXCL14* could maintain the synthesis metabolism of glycerophospholipids and provide the material basis for the development of neurons.

## 4. Materials and Methods

### 4.1. Animals Experiment

C57BL/6n mice (Nihon SLC, Hamamatsu, Japan) were backcrossed with CXCL14^+/−^ mice for more than 10 generations and intercrossed with each other to obtain CXCL14^+/+^, CXCL14^+/−^ and CXCL14^+/+^ littermates. CXCL14^+/+^ male and CXCL14^+/−^ female mice were crossed to produce CXCL14^+/−^ and CXCL14^+/+^ mice. Mice were fed a standard diet (CE-2) (Nihon CLEA, Tokyo, Japan). The mouse model was established by hybridizing 8-week-old mice, segregating them into cages one day after hybridization, and determining pregnancy by observing the abdominal size of female mice. All mice were maintained under a 12 h light, 12 h dark cycle in a pathogen-free animal facility. The Animal Care and Use Committee of Shanghai Jiao Tong University authorized the animal experiments.

### 4.2. Nissl Staining

The parietal cortex of the adult mouse brain was taken for sectioning and the sections were immersed in 0.5% gelatin before being mounted on gelatin-impregnated slides. After defatting the sections, unstained slides were stained with Nissl Differentiation. Finally, slides were dehydrated by sinking into ddH_2_O, 70% ethyl alcohol (twice), 95% ethyl alcohol (twice), 100% ethyl alcohol (twice) and transparentized in xylene (twice) and mounted with neutral balsam. The morphological alterations were observed under a light microscope.

### 4.3. Quantitative RT-PCR

Major organs, including the brain, heart, kidney, liver, lung, small intestine, spleen, stomach, and skin, were collected from 8-week-old male C57 mice. The brain tissue analyzed was from the cerebral cortex. The total RNA of tissues and cell lines was extracted using the RNeasy Plus Universal Tissue Kit (Sangon Biotech Co., Shanghai, China) based on the manufacturer’s instructions. The mRNA was purified as described in Hieff NGS^®^ mRNA Isolation Master Kit mRNA Purification Kit. (Yeasen Biotechnology, Shanghai, China).

From embryonic day 10.5 (E10.5, E12.5, E14.5, E16.5, E18.5), mouse parietal cortex tissues were collected for Quantitative RT-PCR. Primers for CXCL14 and GAPDH were synthesized by Takara (Sangon Biotech Co.). cDNA was constructed using the PrimeScript RT reagent kit (Takara Biotechnology Co., Ltd., Shiga, Japan). Real-time PCR was performed using SYBR premix Ex Taq II (Takara Biotechnology Co., Ltd.) and fluorescence intensity was detected in a Light Cycler 480 system (Roche Applied Science, Sunnyvale, CA, USA) using the following thermocycling conditions: 95 °C for 10 min, 40 cycles of 95 °C for 15 s and 60 °C for 1 min. GAPDH was used as the internal control. U6 and GAPDH were treated as internal controls.

### 4.4. Western Blot

From embryonic day 10.5 (E10.5, E12.5, E14.5, E16.5, E18.5), mouse parietal cortex tissues were collected. The mice were killed by cervical dislocation, the brain was rapidly removed from the skull and the cerebral hemispheres were frozen in isopentane within liquid nitrogen. Samples were stored at −80 °C for subsequent lysis.

Tissue lysates were prepared using RIPA buffer (Beyotime Institute of Biotechnology, Shanghai China). Extracts were mixed with SDS-PAGE loading buffer and denatured by boiling for 5 min. Equal amounts of protein were separated on 10% SDS-PAGE gels and transferred onto polyvinylidene fluoride (PVDF) membranes (Bio-Rad Laboratories, Inc. Hercules, CA, USA). Blots were blocked using 5% bovine serum albumin (BSA, Gibco; Thermo Fisher Scientific, Inc. Waltham, MA, USA) for 2 h and incubated overnight at 4 °C with primary antibodies against CXCL14 (1:1000; ab154390; Abcam, Cambridge, MA, USA) PVDF membranes were washed with Tris-buffered saline (TBS) supplemented with 0.05% Tween-20 before incubation with HRP-conjugated secondary antibody (1:2000; 7074S; Cell Signaling Technology, Inc., Danvers, MA, USA) for 2 h at room temperature. Signals were visualized using enhanced chemiluminescence reagents (Thermo Fisher Scientific, Inc.).

### 4.5. Gene Set Enrichment Analysis

GEPIA2 (http://gepia2.cancer-pku.cn/#index, accessed on 3 December 2023) is an updated version of GEPIA. It can be used to analyze the RNA expression sequencing data of 9736 tumor samples and 8587 normal samples in TCGA and GTEx. The “similar gene detection” module of the GEPIA2 database was used to obtain a gene set that was similar to the CXCL14 expression pattern and ranked in the top 1000. Similar genes detection of function identifies a list of genes with similar expression patterns with CXCL14 and cerebral cortex (for given sets of GTEx expression data). The list of genes that have similar expression patterns ranked by Pearson correlation coefficient (PCC). A standard language with three levels of structure, including biological process (BP), cell composition (CC), and molecular function (MF), has been developed to provide consistent functional descriptions of gene products in various databases. We performed functional enrichment analysis of CXCL14-related genes by using the cluster profile package in R software (topGO) and visualized the results. Through the “correlation analysis” module in the GEPIA2 database, this function performs pair-wise gene expression correlation analysis for given sets of GTEx expression data, using methods including Pearson. One gene can be normalized by other genes.

### 4.6. Membrane-Anchored Ligand and Receptor Yeast Two-Hybrid System

#### 4.6.1. Cell Lines and Plasmids

The E. coli strain DH5α (Code No. 9057) served as the host for plasmid construction and amplification. Yeast strains GoldY2H (MATa, trp1-901, leu2-3, 112, ura3-52, his3-200,gal4Δ,gal80Δ,LYS2::GAL1_UAS_-Gal1_TATA_-His3,GAL2_UAS_-Gal2_TATA_-Ade2 URA3::MEL1_UAS_-Mel1_TATA_ -LacZ, MEL1) and Y187 yeast cells (MATα, ura3-52, his3-200, ade2-101, trp1-901, leu2-3, 112, gal4Δ, met-, gal80Δ, URA3::GAL1_UAS_-GAL1_TATA_-lacZ), were obtained from Clonetech, USA. The plasmids pGAD-T7 and pGBK-T7 (Code No. 630442, 630443) were also acquired from Clonetech, USA. Our laboratory maintained the 293T cell line. The pcDNA3.4 plasmid (Code No. A14697) was purchased from Life Technology, USA. We performed mycoplasma contamination tests on the cells prior to their use.

#### 4.6.2. Construction of Bait Plasmids

The plasmid was synthesized based on previous studies. The pGAD-T7 plasmid was utilized to express the bait, in which all elements between the ADH1 promoter and ADH1 terminator were deleted, with specific target sequences cloned using a one-step clone kit from Vazyme, Co. Nanjing China. The bait protein consisted of a signal peptide, a ligand protein, a transmembrane peptide, the C-terminal region of yeast ubiquitin (*Cub*) and reporter genes. In different experiments, a signal peptide derived from either SP CXCL14 (residues 1–28), SP_WBP1_ (WBP1, residues 1–31) was fused at the N-terminus of ligand The WBP1 transmembrane peptide and flanking sequences (residues 350–430) were used as the transmembrane peptide and linkers and fused between CXCL14 and the reporter genes. To examine subcellular localization, EGFP served as the reporter gene. For detecting PPIs (protein-protein interactions), the Cub-GAL4 cassette functioned as the reporter module.

#### 4.6.3. Construction of Prey Plasmids

Recombinant prey genes were inserted into the pGBK-T7 vector, which involved replacing the original ORF between the ADH1 promoter and terminator with mouse CXC motif receptors and reporter genes. To investigate subcellular localization, EGFP was utilized as a reporter gene, while for PPI detection, *N_ub_G* served as a reporter module.

#### 4.6.4. The Yeast Plasmid Transformation

The mating process followed the LiAc method described in the commercial kit manual. Specifically, prey plasmids were transformed into the Y187 yeast strain, whereas bait plasmids were transformed into the GoldY2H cells strain. Transformants were selected using synthetic dropout nutrient medium/plate (SD medium) lacking leucine or tryptophan (SD/Leu^−^ or SD/Trp^−^). Mating occurred by suspending 2–3 clones of each cell in 0.5 mL YPD medium and incubating overnight with shaking. The resulting cells were collected through centrifugation and spread onto SD/Leu^−^Trp^−^ media plates to facilitate interaction strength detection via growth assay.

#### 4.6.5. Interaction Strength Detection

Growth assay involved dropping five microliters of 10-fold diluted cell suspension (OD_600_ = 1, 0.1 and 0.01 series) onto SD/Leu^−^Trp^−^His^−^Ade^−^ medium plates, followed by photographing the plates after three days of incubation to record cell growth levels accurately.

#### 4.6.6. β-Galactosidase Activity Assay

Hybrid cells (with an OD_600_ ranging from 0.6 to 0.8) were subjected to lysis using Z buffer (composed of 60 mM Na_2_HPO_4_, 40 mM NaH_2_PO_4_, 10 mM KCl, 1 mM MgSO_4_, and adjusted to pH = 7). The lysis process involved three cycles of freezing in liquid nitrogen and thawing in a water bath set at 37 °C. Subsequently, the resulting cell lysates were resuspended in Z buffer containing 0.27% (*v*/*v*) 2-mercaptoethanol and 0.4% 2-nitrophenyl-β-D-galactopyranoside at a temperature of 30 °C, allowing for chromogenic reaction. The reaction was brought to a halt after 40 min by adding 400 μL of 1 M Na_2_CO_3_. The absorbance of the supernatant was then measured at a wavelength of 420 nm. It is important to note that the OD_600_ value corresponds to the absorbance of the yeast culture before lysis. Each sample was subjected to three repetitions of the procedure. The activity of β-Gal was determined using the subsequent equation:(1)$$\beta \text{-galactosidase}\;\mathrm{units}=\frac{{1000\times \mathrm{OD}}_{420}}{{t(\min)\times \mathrm{OD}}_{600}}$$

The test was used as the statistical method.

### 4.7. Metabolomics Analysis

#### 4.7.1. Sample Preparation and Collection

These serum samples were thawed slowly at 4 °C and then 100 μL of each sample was added to 400 μL of precooled methanol/acetonitrile solution (1:1, *v*/*v*), vortexed and mixed, left at −20 °C for 60 min, centrifuged at 14,000× *g* for 20 min at 4 °C and the supernatant was taken and freeze-dried under vacuum. For mass spectrometry, 100 μL of aqueous acetonitrile solution (acetonitrile: water = 1:1, *v*/*v*) was added to redissolve, vortexed, centrifuged at 14,000× *g* at 4 °C for 15 min, and the supernatant was injected into the sample for analysis B at 40%. From 18 to 18.1 min, B changed linearly from 40% to 95%; B was maintained at 95% from 18.1 to 20 min. QC samples were inserted into the sample cohort to monitor and evaluate the stability of the system and the reliability of the experimental data.

#### 4.7.2. LC–MS Method

Each sample was detected by electrospray ionization (ESI) in positive and negative ion modes, respectively. The samples were separated by UPLC and analyzed by mass spectrometry using a Triple-TOF 5600 mass spectrometer (AB SCIEX, Framingham, MA, USA). The ESI Source conditions are as follows: Ion Source Gas1 (Gas1): 60, Ion Source Gas2 (Gas2): 60, Curtain gas (CUR): 30, source temperature: 600 °C, IonSapary Voltage Floating (ISVF) +5500 V (positive and negative modes); TOF MS scan m/z range: 60–1200 Da, product ion scan m/z range: 25–1200 Da, Time of Flight Mass Spectrometer (TOF) MS scan accumulation time 0.15 s/spectra, product ion scan accumulation time 0.03 s/spectra; Secondary mass spectra were obtained with information-dependent acquisition (IDA) and in high sensitivity mode with Declustering potential (DP): +60 V (positive and negative modes), Collision Energy: 30 eV, IDA Settings as follows Exclude isotopes within 4 Da, Candidate ions to monitor per cycle: 6.

#### 4.7.3. Data Analysis

The Content Management System (CMS) program was used for peak alignment, retention time correction and peak area extraction. The structure of metabolites was identified by accurate mass number matching and secondary spectrogram matching to search the laboratory database. For data extracted using LC/MS and GC/MS Data Analysis (XCMS 3.4) software, missing values within groups were eliminated with a specific target of 50% of ion peaks. Application software SIMCA-P 14.1 (Umetrics, Umea, Sweden) pattern recognition was conducted, and the data were preprocessed by Pareto-scaling. After that, multidimensional statistical analysis was performed, including unsupervised principal component analysis (PCA) and orthogonal partial least square discriminant analysis (OPLS-DA). One-dimensional statistics the analysis included Student’s *t*-test and variogram analysis and R software (ggplot2) mapping the volcano.

The MetaboAnalyst 4.0 was applied to identify the most relevant pathway of the differential metabolites by the Kyoto Encyclopedia of Genes and Genomes (KEGG).

## 5. Conclusions

CXCL14 is the evolutionary ancient chemokine for a highly conserved sequence and its homeostatic roles. In this study, it was found that the loss of CXCL14 would lead to abnormal development in mice, especially because the number of neurons in the outer cortical pyramidal layer was significantly reduced. The expression of CXCL14 was highest at embryonic day 14.5 in the embryonic phase and after birth in the mRNA and protein levels. Therefore, we hypothesized that CXCL14 promotes the development of neurons in the somatic layer of the pyramidal cells of mice cortex on embryonic day 14.5. In order to further explore its mechanism, CXCR4 and CXCR7 were suggested as receptors by Membrane-Anchored Ligand and Receptor Yeast Two-Hybrid technology. Through metabolomic techniques, it was inferred that CXCL14 promotes the development of neurons by regulating fatty acid anabolism and glycerophospholipid anabolism. This thesis has provided a deeper insight into the effects of CXCL14 on neuronal development. However, this study did not clarify the link between the CXCR4 and CXCR7 receptors, fatty acid anabolism, and glycerophospholipid anabolism. It is an interesting topic that deserves further research.

## Figures and Tables

**Figure 1 ijms-25-01651-f001:**
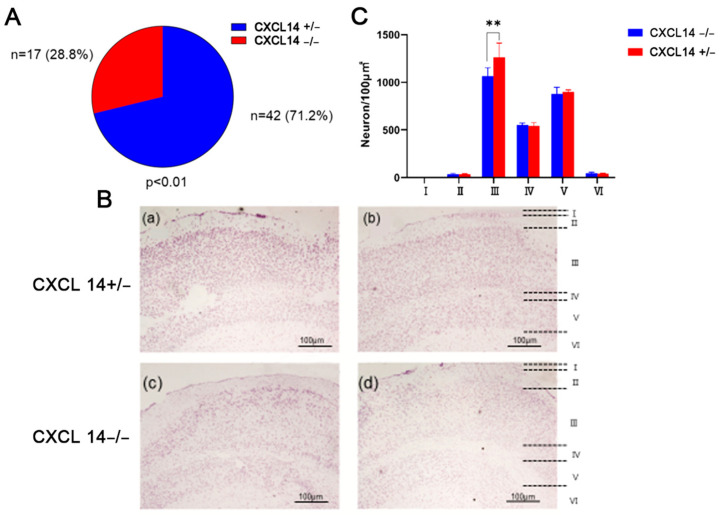
CXCL14 knockout caused the reduction of the number of pyramidal cells in the external pyramidal layer and survival of mice. (**A**) The genotype proportion of offspring born by crossbreeding of 8-week-old male CXCL14^−/−^ and female CXCL14^+/−^ mice, they produced a total of 59 mice, 42 mice of CXCL14^+/−^ and 17 mice of CXCL14^−/−^, and the birth rate of CXCL14^−/−^ was 28.8%, which was significantly different from expected percentage of 50% (*n* = 59, *p* < 0.01). (**B**) Cortical neuron staining of CXCL14^+/−^ and CXCL14^−/−^ 8-week-old mice by Nissl staining. (**a**,**b**) were slice images of CXCL14^+/−^ 8-week-old mice cortexes; (**c**,**d**) were slice image of CXCL14^−/−^ 8-week-old mice cortexes. (Layer I: molecular layer; II: external granular layer; III: external pyramidal layer; IV: internal granular layer; V: external pyramidal layer; VI: Multiform Layer.). (**C**) The density of neurons in each layer of cerebral cortex of CXCL14^−/−^ and CXCL14^+/−^ mice. The neuron density in external pyramidal layer of CXCL14^−/−^ 8-week-old mice was significantly lower than that of their CXCL14^+/−^ sibs. The red line represents CXCL14^−/−^ mice. The black line represents CXCL14^+/−^ mice. (All data are expressed as mean ± SD, *n* = 3, **: *p* < 0.01).

**Figure 2 ijms-25-01651-f002:**
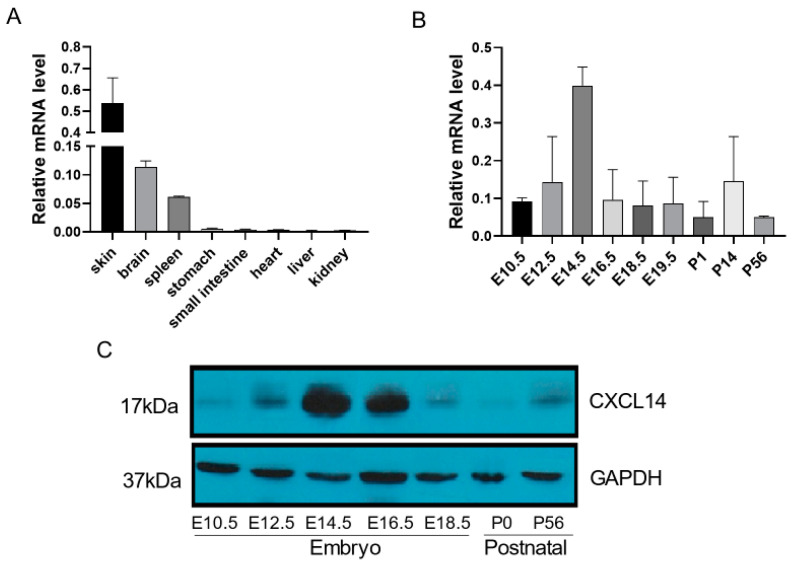
The CXCL14 expression of mice in spatial and temporal. (All data are expressed as mean ± SD, *n* = 3). (**A**) The expression of CXCL14 in major organs of adult male and female 8-week-old mice by Q-PCR, which was higher in the skin and brain than in other organs. (**B**) The expression of CXCL14 by Q-PCR at embryonic days 10.5, 12.5, 14.5, 16.5, 18.5, 19.5 and postnatal days 1, 14, 56. (**C**) The expression of CXCL14 by Western Blot at embryonic days 10.5, 12.5, 14.5, 16.5 18.5 and postnatal days 0, 56.

**Figure 3 ijms-25-01651-f003:**
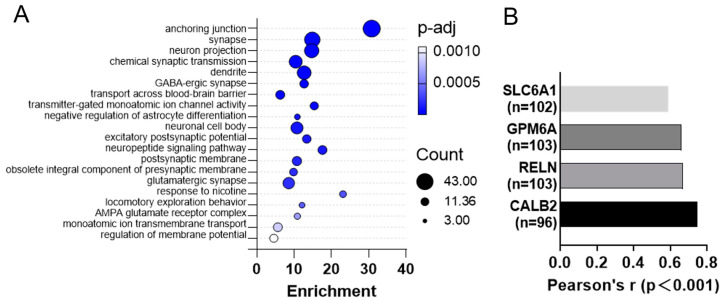
GO enrichment analysis was performed using GEPIA website to analyze the genes with similar expression patterns in the cerebral cortex of *CXCL14* in the GTEx database. (**A**) The top 20 Gene Ontology (GO) enrichment terms in Biological Processes (BP), Cellular Components (CC), and Molecular Functions (MF) in STAD (Stomach Adenocarcinoma) were identified. (**B**) The main genes in the neural projection pathway were positively correlated with *CXCL14*.

**Figure 4 ijms-25-01651-f004:**
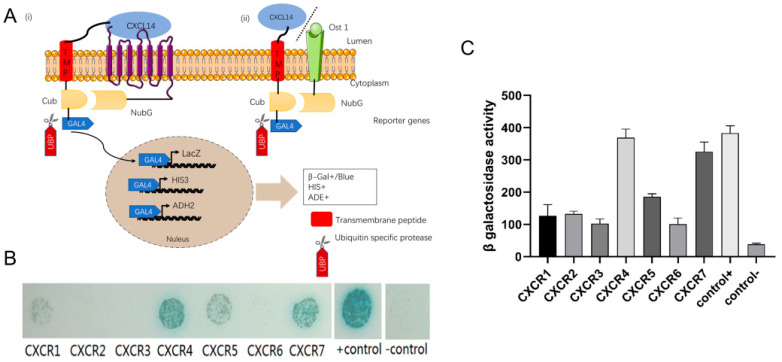
Interaction test of the CXCL14 and CXCR family members. (**A**) Diagram of Membrane-Anchored Ligand and Receptor Yeast Two-Hybrid, MALAR-Y2H. (i) The interaction between CXCL14 and the CXCR series receptor protein was detected. CXCL14 was fused to the C-terminus of ubiquitin protein (*C_ub_*) through a transmembrane peptide (TMP) and conjugated to the transcription factor GAL4. The C-terminus of the receptor protein is coupled to the N-terminus of the ubiquitin protein (*N_ub_*). When CXCL14 interacts with the extracellular domain of the receptor, *C_ub_* and *N_ub_* recombine into active ubiquitin monomers that are hydrolyzed by ubiquitin-specific hydrolase (UBP) to release GAL4. (ii) Negative control, CXCL14 and yeast membrane protein OST1 were used as bait and prey to form a negative control group. (**B**) Yeast Two-hybrid. GoldY2H cells expressing the *CXCL14-CXCL14-TMP-C_ub_-GAL4* plasmid were hybridized with Y187 cells expressing each *OST1-CXCR-N_ub_G* fusion protein, and diploid cells were dropped onto SD/Leu^−^Trp^−^Ade^−^His^−^ plates containing α-X-gal and 15 mM 3-AT. The growth of cells was observed after 72 h cultivation. *CXCL14-TMP-C_ub_-GAL4 × OST1-N_ub_G* was used as a negative control and *WBP1-C_ub_-GAL4 × OST1-N_ub_G* was used as a positive control. (**C**) β-Gal assay of diploid cells expressing *SP-CXCL14-TMP-C_ub_-GAL4* and each of the *SP-OST1-CXCR-N_ub_G* proteins. The experiment was repeated three times independently and every sample was tested twice in each experiment (*n* = 3).

**Figure 5 ijms-25-01651-f005:**
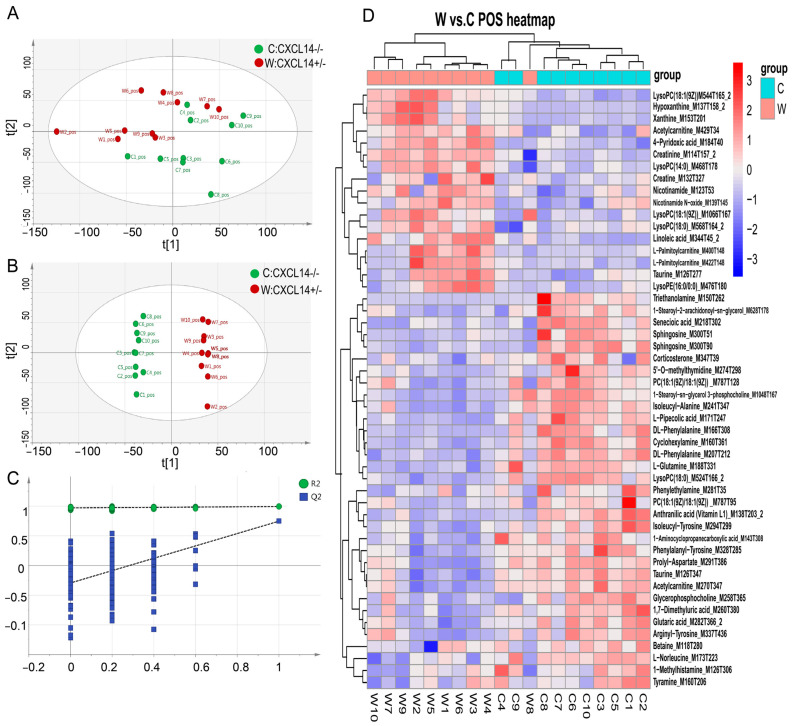
Multivariate data analysis from UPLC-MS/MS. (**A**) PCA score plots of serum samples collected from different groups. (**B**) Score scatter plot of the OPLS-DA model for group CXCL14^+/−^ (W) and CXCL14^−/−^ (C). (**C**) Permutation test of the OPLS-DA model for group CXCL14^+/−^ (W) and CXCL14^−/−^ (C). The abscissa represented the permutation retention of the permutation test and the ordinate represents the value of R2Y or Q2; the green dot represents the value of R2Y obtained by the permutation test and the blue square dot represents the value of Q2 obtained by the permutation test; the two dashed lines represent the regression lines of R2Y and Q2. (**D**) Hierarchical clustering of metabolites with significant differences.

**Figure 6 ijms-25-01651-f006:**
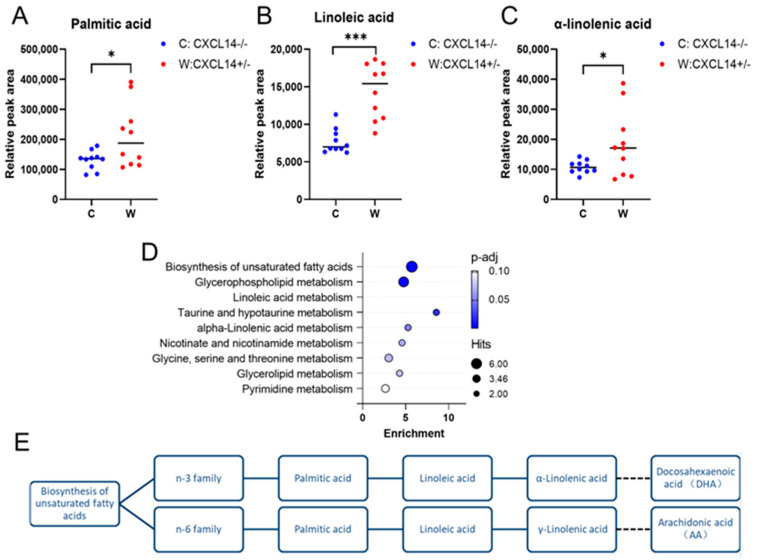
Summary diagram of Met-PA pathway analysis (all data are expressed as mean SD). (**A**) The relative peak area abundance of palmitic acid in CXCL14^−/−^ 8-week-old mice was significantly lower than that in CXCL14^+/−^ 8-week-old mice. (*n* = 10, *: *p* < 0.05). (**B**) The relative peak area abundance of linoleic acid in CXCL14^−/−^ 8-week-old mice was significantly lower than that in CXCL14^+/−^ 8-week-old mice. (*n* = 10, ***: *p* < 0.001). (**C**) The relative peak area abundance of α-linolenic acid in CXCL14^−/−^ 8-week-old mice was significantly lower than that in CXCL14^+/−^ 8-week-old mice. (*n* = 10, *: *p* < 0.05). (**D**) Pathway analysis of differential metabolites for plasma between CXCL14^+/−^ (W) and CXCL14^−/−^ (C) mice by LC-MS metabolomics. (**E**) Differential metabolites in the biosynthetic pathway of unsaturated fatty acid.

## Data Availability

Data is contained within the article and Appendix A.

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
