# Peer review of "CXCL14 as a Key Regulator of Neuronal Development: Insights from Its Receptor and Multi-Omics Analysis"

_ijms, 2024, doi:10.3390/ijms25031651_

Round 1
Reviewer 1 Report
Comments and Suggestions for Authors
The authors of the manuscript titled 'CXCL14 as a Key Regulator of Neuronal Development: Insights from its Receptor and Multi-Omics Analysis' have presented the role of CXCL14 on development of neurons in mice. They showed that CXCL14 could maintain the synthesis metabolism of glycerophospholipids and provide the material basis for the development of neurons. The results are systematically presented and the experiments are easy to follow. Overall, the study appears to be of interest, whereas the experiments have some problems.
Comment #1: The biggest concern with the presented manuscript is that the sample size and age of the presented results are not explained. The authors should clearly state the age and the number of mice used in the presented experiment in the each figure legends.
Fig1 and Fig2 is missing the age and number of mice.
Fig4 described “n=2” in this manuscript, but isn't it correct "n=3"?
The error bar is SD or SE?
Fig6 is missing the age of mice.
Comment #2: Sexual difference
Are there any differences between males and females? There is no clear explanation or description sexual difference of experiments. I hope that the author explain the reason why you do not differentiate between males and females in the revised manuscript.
Comment #3: Discussion and methods
In the line 224, the author described "CXCL14-/+ mice by HE staining (Figure 1B, 1C) However, I think this study is Nissle stain. The authors should modified this point in the revised manuscript.
Comment #4: Abbreviations
Abbreviations used should be defined once the first time they appear in the text.
Ex CXCL, GABA, HE stain and so on
Minor points
Please use appropriate line spacing between figure legends and text in the revised manuscript.
Comments on the Quality of English Language
Please use appropriate line spacing between figure legends and text in the revised manuscript.
Reviewer 2 Report
Comments and Suggestions for Authors
After reviewing the article titled "CXCL14 as a Key Regulator of Neuronal Development: Insights from its Receptor and Multi-Omics Analysis," several suggestions for improvement can be made:
1. The abstract should briefly outline the study's methods and key findings. For instance, mention the specific techniques used in the multi-omics analysis.
2. Introduction: Expand on CXCL14’s role in neurological development. For example, compare its functions with other chemokines in the introduction to set a clear context.
3. State the hypothesis and objectives more explicitly at the end of the introduction, guiding the reader on the study's aims.
4. Provide more details on the selection criteria for animal models and the rationale for choosing specific analytical techniques. This will enhance the reproducibility of the study.
5. Results Presentation: use more descriptive captions for figures to help readers understand their relevance to the study.
6. Discussion Section: Contrast your findings with previous literature in more detail. Discuss how the results align with or differ from existing research, using specific examples from the literature.
7. Explicitly acknowledge any limitations of the study and suggest future research directions. This could involve discussing the applicability of the findings.
8. Update and check all references for relevancy. Include most recent and relevant studies to support findings and discussion points.
Reviewer 3 Report
Comments and Suggestions for Authors
In the manuscript "CXCL14 as a key regulator of neuronal development: insights from its receptor and multi-omics analysis", the authors, Zhang Y, Jin Y, Li J, Yan Y, Wang T, Wang X, Li Z and Qin X. investigate the possible role of chemokine ligand 14 in the embryonal brain development. Zhang et al. crossbred CXCL+/- mice and obtained disproportional low number of CXCL14 -/- mice. They analyzed the neuron density in the layers of cerebral cortex and found it decreased in CXCL14-/- mice. Furtheron, they analyzed the expression of CXCL14 in normal mice in different organs and during embryonal development, on RNA and protein level and found increased expression in the mouse brain on E14.5. They did similar expression pattern gene analysis, as well as specific two hybrid assay to detect CXCL14 interactions with CXCR4 and CXCR7. Finally, by mass spectroscopy and analysis of endogenous metabolites and pathways they found decreased amounts of some fatty acids in CXCL14-/- mice, in comparison with CXCL14+/- mice.
The topic of the manuscript is interesting and not yet widely explored. In the Introduction, more data on the different roles of CLCX14 and its signaling and regulation could be described. In the Results, some data are missing, like how many mice were bred and obtained in comparison with wt mice (some data are given in the figure, but not in the text), how old were mice when their brain was analyzed and how many of these mice were analyzed. It was assumed that abnormal death time of the embryos was E10.5-E14.5, but it is not explained how they proved these facts (except that the CLCX14 expression was high). Also, mass spectroscopy could be better described, with explanations of abbreviations. It would be interesting to have more data on CLCX14 -/- mice: their morphology, behavior, development. The results showed correlation of the CXCL14 expression and fatty acid metabolism, but further experiments should be done to show that CXCL14 is directly regulating metabolism of some of these acids.
Other comments:
Figure legends should give more data on the experimental methods, and not explain the results. Also all abbreviations should be explained. Fonts in some figures (Figure 5) should be bigger. In Figure 3 programs should be mentioned.
Some references have full names of the journals, and some abbreviations. Ref. 2 is listed two times.
Comments on the Quality of English LanguageSentence reorganization: lines 42, 159, 178, 180, 207, 208, 219, 267, 311, 388, 460, 461
Round 2
Reviewer 1 Report
Comments and Suggestions for Authors
I don't have any concerns.
Author Response
Thank the reviewers for his/her encouragement
Reviewer 2 Report
Comments and Suggestions for Authors
The authors responded adequately to my previous comments. As responses satisfy my enquiries, I conclude fore acceptance in the present form.
Author Response
Thank the reviewers for his/her encouragement.
Reviewer 3 Report
Comments and Suggestions for Authors
The authors of the manuscript "CXCL14 as a key regulator of neuronal development: insights from its receptor and multi-omics analysis" responded to some of the comments and improved the manuscript. However, there are still some changes which could be done.
Although some information was added to the Introduction, it seems to me that more basic information about cytokines is given in Discussion than in the Introduction. Considering results, the authors gave the answer how they determined the time of the embryo death in the Cover letter and in supplementary files, but not in the manuscript. So, even if the table remains in the Supplementary files, it could be explained how it was done. In the paragraph describing mass spectroscopy it should be an introductory sentence with explanation about the samples analyzed. Also, on Figures 5 and 6 it should be mentioned which data were analyzed in more details and how were they collected.
Comments on the Quality of English Languagesentence reorganization or corrections: lines 17, 34, 39, 48, 49, 69-72, 111-113, 216, 295, 307, 149, 185, 334, 345, 361, 492, 511.
